# FrugalMCT: Efficient Online ML API Selection for Multi-Label Classification Tasks

## Abstract

Multi-label classification tasks such as OCR and multi-object recognition are a major focus of the growing machine learning as a service industry. While many multi-label APIs are available, it is challenging for users to decide which API to use for their own data and budget, due to the heterogeneity in their prices and performance. Recent work has shown how to efficiently select and combine single-label APIs to optimize performance and cost. However, its computation cost is exponential in the number of labels, and is not suitable for settings like OCR. In this work, we propose FrugalMCT, a principled framework that adaptively selects the APIs to use for different data in an online fashion while respecting the user's budget. It allows combining ML APIs' predictions for any single data point, and selects the best combination based on an accuracy estimator. We run systematic experiments using ML APIs from Google, Microsoft, Amazon, IBM, Tencent, and other providers for tasks including multi-label image classification, scene text recognition and named entity recognition. Across these tasks, FrugalMCT can achieve over 90% cost reduction while matching the accuracy of the best single API, or up to 8% better accuracy while matching the best API's cost.

## 1 Introduction

Many machine learning users are starting to adopt machine learning as a service (MLaaS) APIs to obtain high-quality predictions. One of the most common tasks these APIs target is multi-label classification. For example, one can use Google's computer vision API (Goo) to tag an image with a wide range of possible labels for $0.0015, or Microsoft's API (Mic) for $0.0010. Another example is to extract all text strings from an image for $0.005 via iFLYTEK's API (Ifl) or $0.021 via Tencent's API (Ten). In practice, these APIs also provide different performance on different types of input data (e.g., English vs Chinese text). The heterogeneity in APIs' performance and prices makes it hard for users to decide which API, or combination of APIs, to use for their own datasets and budgets.

Recent work (Chen et al., 2020) proposed FrugalML, an algorithmic framework that adaptively decides which APIs to call for a data point to optimize accuracy and cost. Their approach learns a fast decision rule for each possible output label that can significantly improve cost-performance over the individual APIs. However, FrugalML requires a large amount of training data and involves solving a non-convex optimization problem with complexity exponential in the number of distinct labels. This prevents it from being used for tasks with large number of labels, such as multi-label classification. Furthermore, FrugalML ignores correlation between different APIs' predictions, potentially limiting its accuracy. For example, APIs A and B may output *{person, car}* and *{car, bike}* separately for an image whose true keywords are *{person, car, bike}*. FrugalML would select one of the two label sets, but combining them results in the true label set and thus higher accuracy. Thus, this paper aims to solve these significant limitations and address the question: *how do we design efficient ML API selection strategies for multi-label classification tasks to maximize accuracy within a budget?*

We propose FrugalMCT, a principled framework that learns the strengths and weaknesses of different combinations of multi-label classification APIs, and efficiently selects the optimal combinations of APIs to call for different data items and budget constraints. As shown in Fig. 1 (a), FrugalMCT directly estimates the accuracy of each API combination on a particular input based on the features and predicted labels of that input. Then it uses a fast service selector based on the estimated accuracy to balance accuracy and budget. For example, we might first call API A on an input. If A returns

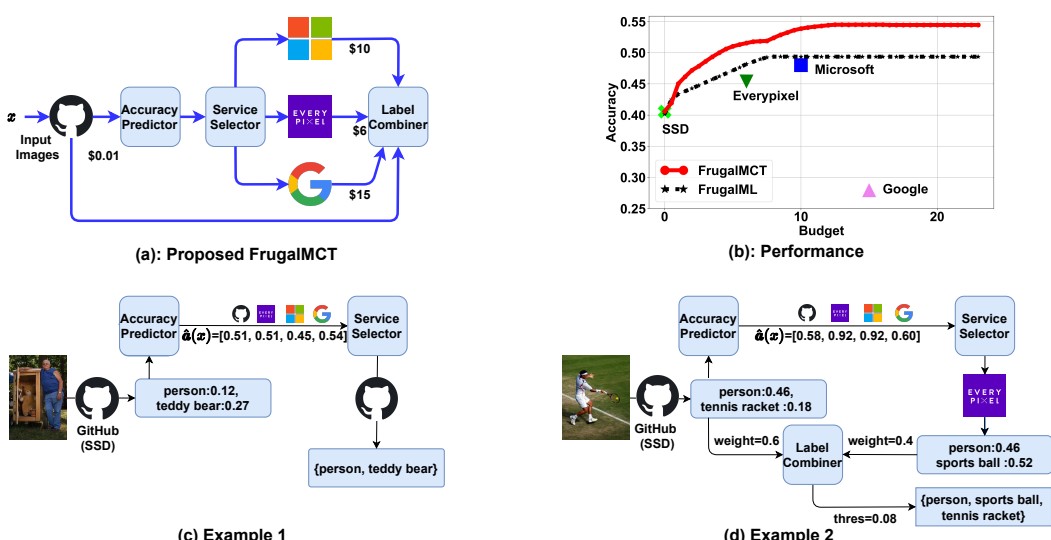

Figure 1: Demonstration of FrugalMCT. **(a)**: FrugalMCT workflow. **(b)**: Performance of FrugalMCT on COCO, a multi-label image dataset, using real commercial ML APIs. **(c), (d)**: Examples of FrugalMCT's behavior on different inputs. In **(c)**, FrugalMCT estimates that the accuracy of a cheap open source SSD model from GitHub is high, and thus directly returns its predictions. In **(d)**, FrugalMCT estimates that combining SSD's results with the Everypixel API has a much higher estimated accuracy, and thus it invokes EveryPixel and combines its results with SSD's results.

*person* and *teddy bear* and the accuracy predictor gives relatively high estimated accuracy (Fig. 1 (c)), then we stop and report *{person, teddy bear}* as the label set. If A returns *person* and *tennis racket*, and we predict that combining it with API B's output gives a much higher accuracy, then we invoke API B and combine their prediction to obtain *{person, sports ball, tennis racket}* (Fig. 1 (d)).

**Contributions.** FrugalMCT is an end-to-end approach that integrates the selection of APIs and the combination of their outputs for individual user queries. It leverages our key finding that current commercial APIs have complementary strengths and weaknesses, and that we can reliably predict which APIs are likely to work well for a new query based on easy-to-generate metadata about its input. FrugalMCT then executes an efficient online algorithm to determine which combination of APIs to call for different user queries. We show that the online algorithm enjoys an accuracy provably close to the offline method as well as a small computational cost. All components in FrugalMCT are trainable, making it easy to customize for different applications. To our knowledge, FrugalMCT is the first work on how to effectively select and combine multi-label ML APIs.

Empirically, FrugalMCT produces substantially better prediction performance than individual APIs and than FrugalML adapted for multi-label tasks (Fig. 1 (b)). Extensive experiments with real commercial APIs on several tasks, including multi-label image classifications, scene text recognition, and named entity recognition, show that FrugalMCT typically provides over 60% (as high as 98%) cost reduction when aiming to match the best commercial API's performance. Also, when targeting the same cost as the best commercial API, FrugalMCT can improve performance up to 8%.

We will release our dataset of 295,212 samples annotated by commercial multi-label APIs as the largest dataset and resource for studying multi-label ML prediction APIs.

**Related work.   MLaaS:** With the growing importance and adoption of MLaaS APIs (Ama; Ten; Goo; IBM; Mic), existing research has largely focused on evaluating individual API for their performance (Yao et al., 2017), robustness (Hosseini et al., 2017), biases (Koenecke et al., 2020) and applications (Buolamwini & Gebru, 2018; Hosseini et al., 2019; Reis et al., 2018). Recent work on FrugalML (Chen et al., 2020) studies API calling strategies for single label classification. While their approach's computational complexity is exponential in the number of labels, FrugalMCT's

complexity does not depend on the number of labels, making it suitable for multi-label prediction APIs. In addition, FrugalML selects only one API per user query, while FrugalMCT considers the combination of multiple APIs' output for each input data. This improves the overall accuracy (as shown in Sec 4), but also creates unique optimization challenges that we solve.

**Ensembles for multi-label classification:** Ensemble learning is a natural approach to combine different predictors' output. Several ensemble methods have been developed, such as using pruned sets (Read et al., 2008), classifier chains (Read et al., 2011), and random subsets (Tsoumakas & Vlahavas, 2007), with applications in image annotations (Xu et al., 2011), document classification (Chen et al., 2017), and speech categorization (Liu et al., 2019). Moyano et al. (2018) provide a detailed survey of this area. Almost all of these ensemble methods require joint training of the base classifiers, but MLaaS APIs are black box to the users. Also, while ensemble methods focus only on improving accuracy, FrugalMCT explicitly considers the cost of each API and enforces a budget constraint.

**Model cascades:** A series of works (Viola & Jones, 2001a;b; Sun et al., 2013; Cai et al., 2015; Wang et al., 2011; Xu et al., 2014; Chen et al., 2018; Kumar et al., 2018; Chen et al., 2018) explores cascades (a sequence of models) to balance the quality and runtime of inference. Model cascades use a *single* predicted quality score to avoid calling computationally expensive models, but FrugalMCT' strategies utilize both *quality scores and predicted label sets* to select an expensive add-on service.

**AutoML for multi-label classification:** AutoML (Thornton et al., 2013) automates the customization of ML pipelines, including the selection, combination, and parametrization of the learning algorithms. There is a rich literature of AutoML techniques for standard single label tasks, and fewer methods on multi-label predictions (Wever et al., 2021) (e.g. genetic algorithms (de Sá et al., 2017) and a neural network-based search scheme (Pakrashi & Namee, 2019)). Applying AutoML to use multiple ML APIs is underexplored, and FrugalMCT can be viewed as the first AutoML approch designed for automating the selection of multiple mutlti-label ML APIs. While most AutoML systems exclusively focus on prediction performance, FrugalMCT optimizes accuracy and cost jointly, which is desirable for cost-sensitive API users.

**Multiple choice knapsack and integer programming:** Many resource allocation problems can be modeled as multiple choice knapsack problem (MCKP) (Pamela H. Vance & Toth), 1993), such as keyword bidding (Zhou & Naroditskiy, 2008) and quality of service control (Lee et al., 1999). While NP-hard (Sinha & Zoltners, 1979), various approximations have been proposed for MCKP, such as branch and bound (Pamela H. Vance & Toth), 1993), convex hull relaxation (Akbar et al., 2006) and bi-objective transformation (Bednarczuk et al., 2018). Inherently an integer linear programming (ILP) problem, MCKP can also be tackled by ILP solvers, motivated by online adwords searching (Devanur & Hayes, 2009), resource allocation (Devanur & Hayes, 2019) and general linear programming (Li et al., 2020). The service selector of FrugalMCT can be viewed as a MCKP with the same item cost vector per item group, which we leverage to obtain a customized fast and online solver.

## 2 PRELIMINARIES

**Notation.** We denote matrices and vectors in bold, and scalars, sets, and functions in standard script. Given a matrix $\mathbf{A} \in \mathbb{R}^{n \times m}$, we let $\mathbf{A}_{i,j}$ denote its entry at location $(i, j)$. $\mathbb{1}(\cdot)$ represents the indicator function.

**Multi-label classification Tasks.** Throughout this paper, we focus on multi-label classification tasks: assigning a label set $Y \subseteq \mathcal{Y}$ to any data point $x \in \mathcal{X}$. In contrast to basic supervised learning, in multi-label learning each data point is associated with a set of labels instead of a single label. Many MLaaS APIs target such tasks. Consider, for example, image tagging, where $\mathcal{X}$ is a set of images and $\mathcal{Y}$ is the set of all tags. Example label sets could be {*person*, *car*} or {*bag*, *train*, *sky*}.

**MLaaS Market.** Consider a MLaaS market consisting of $K$ different ML services for some multi-label task. For a data point $x$, the $k$th service returns to the user a set of labels with their quality scores, denoted by $Y_k(x) \subseteq \mathcal{Y} \times [0, 1]$. For example, one API for multi-label image classification might produce $Y_k(x) = \{(person, 0.8), (car, 0.7)\}$, indicating the label *person* with confidence 0.8 and *car* with confidence 0.7. Let the vector $\boldsymbol{c} \in \mathbb{R}^K$ denote the unit cost of all services. For example, $\mathbf{c}_k = 0.01$ means that users need to pay \$0.01 every time they call the $k$th service.

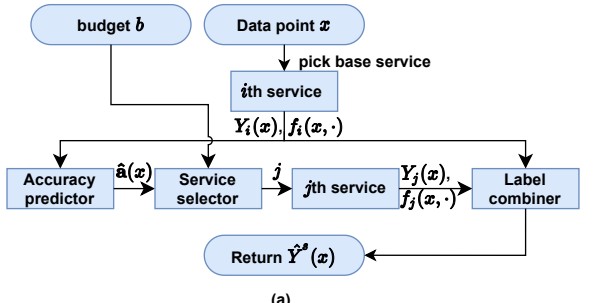

| Symbol | Meaning |
|---|---|
| $x$ | data point |
| $b \in \mathbb{R}^+$ | budget |
| $i, j$ | service indexes |
| $Y_i(x) \subseteq \mathcal{Y}$ | $i$th service's labels |
| $f_i(x, \cdot) : Y_i(x) \mapsto [0, 1]$ | quality score function |
| $\hat{a}(x) \in \mathbb{R}^K$ | estimated accuracy |
| $\hat{Y}^s(x) \subseteq \mathcal{Y}$ | returned labels |

(b)

Figure 2: Overview of FrugalMCT. **(a)** shows how it works: Given a data point, FrugalMCT first invokes a base service. An accuracy predictor then estimates the performance of different APIs. Next, an add-on service is selected based on the predicted accuracy and budget. Finally, the add-on and base services' predictions are combined to return FrugalMCT's prediction. **(b)** lists notation.

## 3 FRUGALMCT FRAMEWORK

In this section, we present FrugalMCT, a framework to adaptively select ML APIs for multi-label classification tasks within a budget. All proofs are left to the appendix. We generalize the scheme in Figure 1 (a) to $K$ ML services. As shown in Figure 2, FrugalMCT contains three main components: an accuracy estimator, a service selector, and a label combiner.

Given a data point $x$, it first calls some base service, denoted by *base*, which is one of the $K$ APIs, and obtains $Y_{base}(x)$. Often, *base* is a cheap or free service; we discuss how to choose *base* in Section 3.4. Next, an accuracy predictor produces a vector $\hat{a}(x) \in [0, 1]^K$, whose $k$th value estimates the accuracy of the label set produced by the label combiner using base's and $k$th API's outputs. The service selector $s(\cdot) : \mathcal{X} \mapsto [K]$ then decides if and which *add-on service* needs to be invoked. Finally, a label combiner generates a label set by combining the predictions from the base and add-on APIs. Take Figure 1 (d) as an example. The image is first passed to the GitHub model, which produces {(*person*, 0.46),(*tennis racket*,0.18)}, by which the accuracy predictor predicts the accuracy of the label set generated by combining each API's output with GitHub model's. The service selector then decides to further invoke Everypixel, which gives {(*person*, 0.46), (*sports ball*, 0.52)}. Finally, the label combiner uses both APIs' output for the final prediction.

FrugalMCT allows users to customize the accuracy predictor and the label combiner, depending on the applications. For example, for the image tagging problem, one might use image features (e.g., brightness and contrast) to build the accuracy predictor, while word embeddings can be more useful for named entity recognition. In the following sections, we explain the key of accuracy predictor, API selector and the label combiner in more detail.

### 3.1 ACCURACY PREDICTION

The accuracy predictor $\hat{a}(\cdot)$ can be obtained by two steps. The first step is to generate a feature vector for every data point in the training dataset $\mathbb{X}^{Tr} \triangleq \{x_1^{Tr}, x_2^{Tr}, \cdots, x_{N^{Tr}}^{Tr}\}$. Generally the feature vector can be any embedding of the data point $x$ and base service prediction $Y_{base}(x)$. In this paper we adopt a simple approach: if the label set $\mathcal{Y}$ is bounded, a $|\mathcal{Y}|$ dimensional vector is generated using one hot encoding on $Y_{base}(x)$. For example, given $\mathcal{Y} = \{person, car, bike\}$ and $Y_{base}(x) = \{(person, 0.8), (car, 0.7)\}$, the generated feature vector is $[0.8, 0.7, 0]$. For unbounded $\mathcal{Y}$, word embedding is used to generate a vector for every predicted label, and the sum of them (weighted by their quality values) becomes the corresponding feature vector.

The next step is to train the accuracy predictor. For each $x_n^{Tr} \in \mathbb{X}^{Tr}$, as its true label sets and prediction from each API are available, we can construct its true accuracy vector $a(x_n^{Tr}) \in [0, 1]^K$, whose $k$th element is the accuracy of the label produced by the label combiner using base and $k$th service predictions. Then we can train some regressor (e.g., random forest) to map the feature vector to the accuracy vector. We use standard multi-label accuracy[1] (Zhang & Zhou, 2014) as a concrete

---

[1] $\frac{\|Y \cap Y'\|}{\|Y \cup Y'\|}$ where $Y/Y'$ is the true/predicted label set.

metric. FrugalMCT can as easily use another metric such as F1-score, precision or subset accuracy.

## 3.2 THE API SELECTION PROBLEM

A key part of FrugalMCT is the API selector $s$: given a budget $b$ and the estimated accuracy $\hat{\boldsymbol{a}}(x)$, which service should be invoked? Let $\mathbb{X} \triangleq \{x_1, x_2, \cdots, x_N\}$ be the entire unlabeled dataset to be classified, and $S \triangleq \{1, 2, \cdots, K\}^{\mathbb{X}}$ be the set of all functions mapping each data point in $\mathbb{X}$ to an API. Let *base* be the index of the base service. For any $s \in S$, $s(x) = base$ implies no add-on API is needed, and $s(x) = k \neq base$ implies $k$th API is invoked. Our goal is to find some $s \in S$ to maximize the estimated accuracy while satisfying the budget constraint, formally stated as below.

**Definition 1.** *Let $\boldsymbol{Z}^*_{n,k}$ be the optimal solution to the budget aware API selection problem*

$$\max_{\boldsymbol{Z} \in \mathbb{R}^{N \times K}:} \frac{1}{N} \sum_{n=1}^{N} \sum_{k=1}^{K} \boldsymbol{Z}_{n,k} \hat{\boldsymbol{a}}_k(x_n)$$

$$s.t. \frac{1}{N} \sum_{n=1}^{N} \sum_{k=1, k \neq base}^{K} \boldsymbol{Z}_{n,k} \boldsymbol{c}_k + \boldsymbol{c}_{base} \leq b; \sum_{k=1}^{K} \boldsymbol{Z}_{n,k} = 1, \forall n; \boldsymbol{Z}_{n,k} \in \{0, 1\}, \forall n, k \tag{3.1}$$

*Then the optimal FrugalMCT strategy is given by $s^*(x_n) \triangleq \arg\max_k \boldsymbol{Z}^*_{n,k}$.*

Here, the objective quantifies the average accuracy, the first constraint models the budget requirement, and the last two constraints enforces only one add-on API is picked for each data point. Base service is needed for every data point and thus its cost $\boldsymbol{c}_{base}$ appears for every $n$ in the budget constraint. Note that Problem 3.1 is a MCKP (and thus integer linear program) and NP-hard in general.

## 3.3 AN ONLINE ALGORITHM FOR FRUGALMCT

In many time-sensitive applications, the input data $x_n$ (as well as the accuracy vector $\hat{\boldsymbol{a}}(x_n)$) comes sequentially, and the API needs to be selected before observing the future data. The selection process also needs to be fast.

To tackle this challenge, we present an efficient online algorithm, which requires $O(K)$ computations per round and gives a provably near-optimal solution. The key idea is to explicitly balance between accuracy and cost at every iteration. Specifically, for a given data point $x_n$ and $p \in \mathbb{R}$, let us define a strategy $s^p(x_n) \triangleq \arg\max_k \hat{\boldsymbol{a}}_k(x_n) - p\boldsymbol{c}_k \mathbb{1}_{k \neq base}$ and break ties by picking $k$ with smallest cost. Here, $p$ is a parameter to balance between accuracy $\hat{\boldsymbol{a}}(x_n)$ and cost $\boldsymbol{c}$. When $p = 0$, $s^p(x_n)$ selects the API with highest estimated accuracy. When $p$ is large enough $s^p(x_n)$ enforces to pick the base API. In fact, larger value of $p$ implies more weights on cost and smaller $p$ favors more the accuracy. Let $r(s) \triangleq \frac{1}{N} \sum_{n=1}^{N} \hat{\boldsymbol{a}}_{s(x_n)}(x_n)$ denote the average accuracy achieved by a strategy $s$. We can show, interestingly, an appropriate choice of $p$ leads to small average accuracy loss.

**Theorem 1.** *Assume the probability density of $\hat{\boldsymbol{a}}(x)$ is a continuous function on $[0, 1]^K$. Then with probability 1, there exists $p^*$ such that $s^{p^*}$ satisfies budget constraint, and $r(s^{p^*}) \geq r(s^*) - \frac{1}{N}$.*

In words, $s^{p^*}(x_n)$ gives a solution to the API selection problem with accuracy loss at most $\frac{1}{N}$. In practice, $\hat{\boldsymbol{a}}(x)$ is continuous for standard ML models of accuracy predictors (e.g., logistic regressors) and thus the assumption holds. In addition, it is computationally efficient: at iteration $n$, it only requires computing $\hat{\boldsymbol{a}}_k(x_n) - p\boldsymbol{c}_k \mathbb{1}_{k \neq base}$ for $k = 1, 2, \cdots, K$, which takes only $O(K)$ computations.

The remaining question is how to obtain $p^*$. As we cannot see the future data to compute $p^*$, a natural idea is to estimate it using the training dataset. More precisely, given the training dataset $\{x_1^{Tr}, x_2^{Tr}, \cdots, x_{N^{Tr}}^{Tr}\}$, let $\hat{p}, \hat{\boldsymbol{q}}$ be the optimal solution to the following problem

$$\min_{p, \boldsymbol{q} \geq 0} (1 - \delta)(b - \boldsymbol{c}_{base})p + \sum_{n=1}^{N^{Tr}} \boldsymbol{q}_n, \ s.t. \frac{\boldsymbol{c}_k \cdot \mathbb{1}_{k \neq base} \cdot p}{N^{Tr}} + \boldsymbol{q}_n \geq \frac{\hat{\boldsymbol{a}}_k(x_n^{Tr})}{N^{Tr}}, \forall n, k \tag{3.2}$$

where $\delta \in (0, 1)$ is a small buffer to ensure that we don't exceed the budget (in practice we set $\delta \leq 0.01$). Technically, Problem 3.2 is the dual problem to the linear programming by relaxing the

integer constraint in Problem 3.1 on the training dataset with budget $(1 - \delta)b$, and $\hat{p}$ corresponds to the near-optimal strategy for the training dataset. If the training and testing datasets are from the same distribution, then a small $\delta$ can ensure with high probability, $\hat{p}$ is slightly less than $p^*$ and thus $s^{\hat{p}}$ satisfies the budget constraint. Given $\hat{p}$, one can use $s^{\hat{p}}$ to select the APIs in an online fashion. The details are given in Algorithm 1.

---

**Algorithm 1** FrugalMCT Online API Selection Algorithm.

---

**Input** : $\boldsymbol{c}, b, \{x_1^{Tr}, x_2^{Tr}, \cdots, x_{N^{Tr}}^{Tr}\}, \{x_1, x_2, \cdots, x_N\}$
**Output** : FrugalMCT online API selector $s^o(\cdot)$
1: Compute $\hat{p}$ by solving Problem 3.2 and set $b^r = N(b - \boldsymbol{c}_{base})$.
2: At iteration $n = 1, 2, \cdots, N$:
3: $s^o(x_n) = \begin{cases} s^{\hat{p}}(x_n) & \text{if } b^r - \boldsymbol{c}_{s^{\hat{p}}(x_n)} \geq 0 \\ base & o/w \end{cases}$
4: $b^r = b^r - \boldsymbol{c}_{s^{\hat{p}}(x_n)} \mathbb{1}_{s^{\hat{p}}(x_n) \neq base}$

---

Here, $b^r$ is used to ensure the generated solution is always feasible. The following theorem gives the performance guarantee of the online solution.

**Theorem 2.** *If* $\delta = \Theta\left(\sqrt{\frac{\log N/\epsilon}{N}} + \sqrt{\frac{\log N^{Tr}/\epsilon}{N^{Tr}}}\right)$ *and the probability density of* $\hat{\boldsymbol{a}}(x)$ *is a continuous function on* $[0, 1]^K$*, then* $s^o$ *satisfies the budget constraint, and with probability at least* $1 - \epsilon$*,* $r(s^o) \geq r(s^*) - O\left(\sqrt{\frac{\log N/\epsilon}{N}} + \sqrt{\frac{\log N^{Tr}/\epsilon}{N^{Tr}}}\right)$.

Roughly speaking, $s^o$ leads to an accuracy loss at most $O\left(\sqrt{\frac{\log N}{N}} + \sqrt{\frac{\log N^{Tr}}{N^{Tr}}}\right)$ compared to the optimal offline strategy. For large training and testing datasets, such an accuracy loss is often negligible, which is also verified by our experiments on real world datasets.

### 3.4 BASE SERVICE SELECTION AND LABEL COMBINATION

Now we describe how the base service is selected and how the label combiner works. The base service can be picked by an offline searching process. More precisely, for each possible base service, we train a FrugalMCT strategy and evaluate its performance on a validation dataset, and pick the base service corresponding to the highest performance.

The label combiner contains two phases. First, a new label set associated with its quality function is produced. The label set is simply the union of that from the base service and add-on service. The quality score is a weighted sum of the score from both APIs, controlled by $w$. For example, suppose the base predicts $\{(person, 0.8), (car, 0.7)\}$ and the add-on predicts $\{(car, 0.5), (bike, 0.4)\}$. Given $w = 0.3$, new confidence for *person* is $0.3 \times 0.8 = 0.24$, for *car* is $0.3 \times 0.7 + 0.7 \times 0.5 = 0.46$, and for *bike* is $0.7 \times 0.4 = 0.28$. Thus the combined set is $\{(person, 0.24), (car, 0.46), (bike, 0.28)\}$. Next, a threshold $\theta$ is applied to remove labels with low confidence. For example, given $\theta = 0.25$, the label *person* would be removed, and the final predicted label set becomes $\{car, bike\}$. The parameters $w$ and $\theta$ are global hyperparameters for each dataset, and can be obtained by an efficient searching algorithm to maximize the overall performance. The details are left to Appendix A.

## 4 EXPERIMENTS

We compare the accuracy and incurred costs of FrugalMCT to that of real world ML services for various tasks. Our goal is to (i) understand when and why FrugalMCT can reduce cost without hurting accuracy, (ii) investigate the trade-offs between accuracy and cost achieved by FrugalMCT, and (iii) assess the effect of training data size and accuracy predictors on FrugalMCT's performance.

**Tasks, ML services, and Datasets.** We focus on three common ML tasks in different application domains: multi-label image classification (*MIC*), scene text recognition (*STR*), and named entity recognition (*NER*). *MIC* aims at obtaining all keywords associated with an image, *STR* seeks to

Table 1: ML services used for each task. Price unit: USD/10,000 queries. A publicly available (and thus free) GitHub model is also used per task: a single shot detector (SSD) (SSD) pretrained on Open Images V4 (Kuznetsova et al., 2020) for *MIC*, a convolutional recurrent neural network (PP-OCR) (Pad) pretrained on an industrial dataset (Du et al., 2020) for *STR*, and a convolutional neural network (spaCy (Spa)) pretrained on OntoNotes (Weischedel et al., 2017) for *NER*.

| Task | ML Service | Price | ML Service | Price | ML Service | Price |
|------|-----------|-------|-----------|-------|-----------|-------|
| MIC | Everypixel (Eve) | 6 | Microsoft (Mic) | 10 | Google (Goo) | 15 |
| STR | Google (Goo) | 15 | iFLYTEK (Ifl) | 50 | Tencent (Ten) | 210 |
| NER | Amazon (Ama) | 3 | Google (GoN) | 10 | IBM (IBM) | 30 |

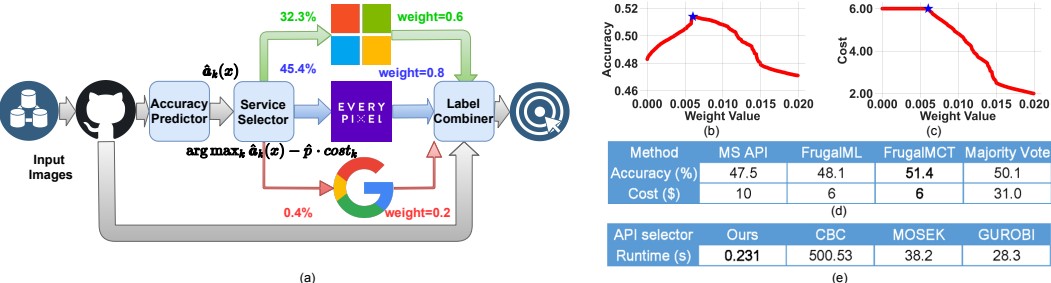

Figure 3: A FrugalMCT strategy learned on the dataset COCO. **(a)** shows that FrugalMCT reduces cost by mostly calling the Everypixel API (45.4%) or the GitHub API (22.1%) only. **(b)** and **(c)** show how the accuracy and cost vary with weight $p$. The blue point corresponds to $0.006$, the learned $\hat{p}$. (d) shows the accuracy and cost of FrugalMCT, FrugalML, Microsoft API, and majority vote. (e) gives the runtime performance of our (online) API selector and three commercial ILP solvers.

recognize all texts in an image, and *NER* desires to extract all entities in a text paragraph. The ML services used for each task as well as their prices are summarized in Table 1. For each task we use three datasets, with details in Appendix C.

**Accuracy Predictors.** Except when explicitly noted, we use a random forest regressor as the accuracy predictor for all the datasets. For *MIC* and *STR* datasets, we map each possible label to an index, and create a feature vector whose $k$th element is base service's quality score for the label corresponding to $k$. If a label is not predicted, the corresponding value is 0. For *NER* datasets, we map each predicted label to a 96-dimensional vector using a word embedding from spaCy (Spa), and then use the sum weighted by their corresponding quality scores as the feature vector. The accuracy predictor is then trained on half of the datasets using the feature vectors generated as above.

**Multi-label Image Classification: A Case Study.** Let us start with multi-label image classification on the COCO dataset. We set budget $b = 6$, the price of Everypixel, the cheapest commercial API (except the open source model from GitHub). For comparison, we also use the average quality score over all predicted labels as the confidence score and adapt FrugalML (Chen et al., 2020) with the same budget ($= 6$) as another baseline .

Figure 3 demonstrates the learned FrugalMCT strategy. As shown in Figure 3 (a), the learned FrugalMCT reduces the cost by mostly using the Everypixel API (45%, 6$) and occasionally calling Microsoft API (32%, 10$).Note that its performance depends on the threshold value $\hat{p}$. As shown in Figure 3 (b) and (c), for small thresholds, FrugalMCT tends to call the more accurate and expensive APIs. However, it runs out of budget quickly, and for many data points only base service can be used, leading to low accuracy. For large thresholds, FrugalMCT tends to call cheaper but less accurate APIs, failing to fully use the budget and thus causing low accuracy too. The $\hat{p}$ value learned by FrugalMCT (blue point in Figure 3 (b) and (c)) produces the optimal accuracy given the budget. Figure 3 (d) shows that FrugalMCT's accuracy (0.514) is higher than that of the best ML service (MS, 0.475) and majority vote (Maj 0.501), while its cost is much lower. This is primarily because

Table 2: Cost savings achieved by FrugalMCT that reaches same accuracy as the best commercial API. On average the cost saving across the evaluated datasets is 73%.

| Dataset | Accuracy (%) | Best API $ | Our $ | Save |
|---|---|---|---|---|
| PASCAL (Everingham et al., 2015) | 74.8 | 10 | 1.4 | 86% |
| MIR (Huiskes & Lew, 2008) | 41.2 | 10 | 4.2 | 58% |
| COCO (Lin et al., 2014) | 47.5 | 10 | 3 | 70% |
| MTWI (He et al., 2018) | 67.9 | 210 | 30 | 86% |
| ReCTS (Zhang et al., 2019) | 61.3 | 210 | 78 | 63% |
| LSVT (Sun et al., 2019) | 53.8 | 210 | 67 | 68% |
| CONLL (Sang & Meulder, 2003) | 52.6 | 3 | 1.5 | 50% |
| ZHNER (ZHN) | 61.3 | 30 | 0.7 | 98% |
| GMB (Bos, 2013) | 50.1 | 30 | 4.1 | 80% |

FrugalMCT learns when the cheaper APIs perform better and call them aptly. FrugalMCT also outperforms FrugalML by exploiting the label combination. As shown in Figure 3 (e), the API selector of FrugalMCT (Alg. 1) is several orders of magnitude faster than commercial ILP solvers, by leveraging the specific structure of Problem 3.1.

**Analysis of Cost Savings.** Next, we evaluate how much cost can be saved by FrugalMCT to reach the highest accuracy produced by a single API on different tasks. As shown in Table 2, FrugalMCT can typically save more than 60% of the cost. Interestingly, the cost saving can be up to 98% on the dataset ZHNER. This is probably because (i) the accuracy estimator enables the API selector to identify when the base service's prediction is reliable and to avoid unnecessarily calling add-on services, and (ii) when add-on API is invoked, the apt combination of the base and add-on services leads to a high accuracy improvement.

**Accuracy and Cost Trade-offs.** Now we dive deeply into the accuracy and cost trade-offs achieved by FrugalMCT, shown in Figure 4. We compare with two oblations: "Offline", where the full data is observed before making decision, "DAP", where a dummy accuracy predictor is used, which, for each API, always returns its mean accuracy on the training dataset. We also compared with an adapted version of the previous state-of-the-art for single label task, FrugalML. To adapt it to multi-label tasks, we use the average quality score over all predicted labels as a single score, and cluster all labels into a "superclass".

Compared to any single API, FrugalMCT allows users to pick any point in its trade-off curve and offers substantial more flexibility. In addition, FrugalMCT often achieves higher accuracy than any ML services it calls. For example, on COCO and ZHNER, more than 5% accuracy improvement can be reached with the same cost of the best API. Note that FrugalMCT also outperforms FrugalML with the same budget. This is primarily because FrugalMCT (i) utilizes a more principled way to use the features (learning an accuracy estimator) than FrugalML (directly using the label info), and (ii) adopts a label combiner designed for multi-label tasks. Ensemble methods such as majority votes (in the appendix C) produce accuracy similar to FrugalMCT, but their cost is much higher.

Note that there is little performance difference between the online FrugalMCT strategy and the offline approach, due to the carefully designed online algorithm. This directly supports our theory. The accuracy predictors play an important role in FrugalMCT's performance. As Table 4 shows, FrugalMCT is able to provide nontrivial accuracy estimates which enables its success. It's interesting to note that the accuracy predictor doesn't need to be perfect for FrugalMCT to do well; for example, the root mean square error (RMSE) of the accuracy predictor is 0.28 on PASCAL, but FrugalMCT still produces consistently better accuracy than FrugalML. We also evaluated FrugalMCT's performance when the accuracy predictors are obtained via two AutoML toolkits, auto-sklearn (Feurer et al., 2015) and Auto-PyTorch (Mendoza et al., 2019) instead of random forest, and observe a similar performance.

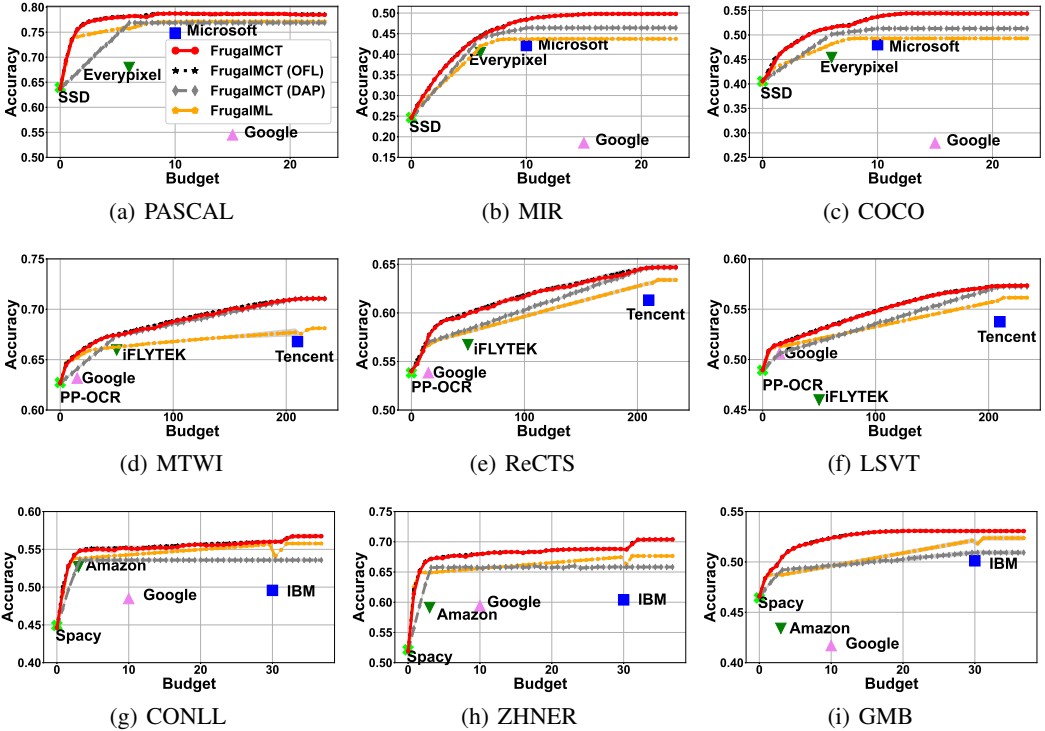

Figure 4: Accuracy cost trade-offs. The offline FrugalMCT (black) observes the full data and then make decisions. The online FrugalMCT (red) matches the offline performance in all the experiments. DAP (grey) is an oblation of FrugalMCT where a dummy accuracy predictor is used. FrugalML (orange) is the previous state-of-the-art method. The task of row 1, 2, 3 is *MIC*, *STR*, and *NER*.

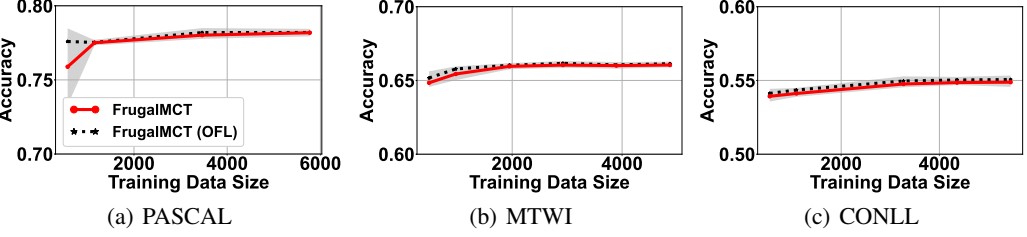

Figure 5: Testing accuracy v.s.training data size. The fixed budget is 6, 15, 3, respectively.

**Effects of Training Sample Size.** Finally we study how the training dataset size affects FrugalMCT's performance. As shown in Figure 5, across different tasks, a few thousand training samples are typically sufficient to learn the optimal FrugalMCT strategy. This is usually more efficient than training a customized ML model from scratch.

## 5 CONCLUSION

In this paper, we presented FrugalMCT, an algorithmic framework to adaptively select and combine ML APIs for multi-label classification tasks within a budget constraint. FrugalMCT integrates forecasts of API's accuracy with online constrained optimization to create an end-to-end algorithm with strong empirical performance and theoretical guarantees. How to efficiently use multi-label APIs is an important problem in practice for the large number of ML users who have chosen to rely on commercial prediction APIs, and has not been studied heavily in the ML literature. This work can help MLaaS users improve both the overall accuracy and cost of their applications. Extensive empirical evaluation using real commercial APIs shows that FrugalMCT significantly improves both cost and accuracy. To encourage more research on MLaaS, we also release the dataset used to develop FrugalMCT, consisting of 295,212 samples annotated by commercial multi-label prediction APIs.

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
