# OpenReview forum: "FrugalMCT: Efficient Online ML API Selection for Multi-Label Classification Tasks"
_ICLR.cc/2022/Conference — ICLR 2022 Submitted_

### Official Review · Reviewer_S7of · 2021-11-02

**Correctness:** 2
**Technical Novelty And Significance:** 1
**Empirical Novelty And Significance:** 1
**Recommendation:** 3
**Confidence:** 4

**Main Review:**

The paper amis to porpose a novel MLC method within a budget limitation. The technique is very old, the novelty and the experiments are very limited.
1.The paper claims that "FrugalML requires a large amount of training data and involves solving a non-convex optimization problem with complexity exponential in the number of distinct labels". However, the paper does not provide the time cost of their proposed method and also do not report their time cost in the experiments. Since the paper aims to design an efficient ML API selection strategies for multi-label classification tasks, but the paper did no show any benifit of time of their proposed method from both thoery and experiments. This is a major flaw.
2,The paper just combines existing technique, but did not bring any new technique, or new theory or new insight.
3.The paper presents online algorithm. Then what is the regret bounds, i.e. bounding the regret of the proposed algorithm over the worst-case sequence?
4.There are too many efficient XMLC methods. What are the advantanges and disadvantages of the proposed methods and other existing MLC or XMLC methods?What is the different and connection between them?
5.Can the proposed method converge? Can you provide the theory and experiments guarantee?
6.Can you test your method on XMLC datasets? Can you compare your method with popular XMLC methods such as PPD-Sparse, SLEEC,SiameseXML,ECLARE,LightXML on Amazon-3M and WikiLSHTC-325K	? please refer to http://manikvarma.org/downloads/XC/XMLRepository.html

**Summary Of The Paper:**

The paper porposes a method to adaptively select and combine ML APIs for multi-label classificaiton within a budget limitation. The techinique is very old, the novelty and the experiments are very limited.

**Summary Of The Review:**

1.The paper claims that "FrugalML requires a large amount of training data and involves solving a non-convex optimization problem with complexity exponential in the number of distinct labels". However, the paper does not provide the time cost of their proposed method and also do not report their time cost in the experiments. Since the paper aims to design an efficient ML API selection strategies for multi-label classification tasks, but the paper did no show any benifit of time of their proposed method from both thoery and experiments. This is a major flaw.
2,The paper just combines existing technique, but did not bring any new technique, or new theory or new insight.
3.The paper presents online algorithm. Then what is the regret bounds, i.e. bounding the regret of the proposed algorithm over the worst-case sequence?
4.There are too many efficient XMLC methods. What are the advantanges and disadvantages of the proposed methods and other existing MLC or XMLC methods?What is the different and connection between them?
5.Can the proposed method converge? Can you provide the theory and experiments guarantee?
6.Can you test your method on XMLC datasets? Can you compare your method with popular XMLC methods such as PPD-Sparse, SLEEC,SiameseXML,ECLARE,LightXML on Amazon-3M and WikiLSHTC-325K	? please refer to http://manikvarma.org/downloads/XC/XMLRepository.html

---

> ### Author Response · Authors · 2021-11-18
> **Clarification of this paper: no major flaw**
>
> Thank you for your feedback. We answer your questions as follows.
>
> ***[The paper aims to propose a novel MLC method]***: We want to stress that this is NOT the paper’s aim. Instead, the goal is to automatically determine which commercial ML APIs to call for ML-as-a-service (MLaaS) users with different budget constraints. This is an under-explored area in research, as the budget requirements and black-box nature of commercial ML APIs brings unique challenges and calls for new techniques such as FrugalMCT. It is also practically relevant to the many ML users who rely on commercial APIs. To the best of our knowledge, FrugalMCT is the first method that adaptively selects multi-label ML APIs to reduce invocation cost and improve performance. Our findings are validated across nine tasks using real-world commercial ML APIs.
>
> ***[What is the time cost of the proposed method?]***:  Figure 3(e) gives the time cost of the proposed online API selector (0.23s), which is orders of magnitude faster than commercial ILP solvers. The end-to-end inference and training times are 0.47s and 59.5s respectively.
>
> ***[What is the new technique and insight?]***: The new technique is the proposed framework, FrugalMCT, an end-to-end approach that integrates the selection of ML APIs and the combination of their outputs for individual user queries. It leverages our key insight that current commercial APIs have complementary strengths and weaknesses, and that we can reliably predict which APIs are likely to work well for a new query based on easy-to-generate metadata about its input. We also design an efficient online algorithm to determine which combination of APIs to call for different user queries. We show that the online algorithm enjoys an accuracy that is provably close to the offline method as well as a small computational cost.
>
> ***[What is the regret bound for the online algorithm?]***: Please refer to Theorem 2 for the performance guarantee. Roughly speaking, the accuracy loss due to the online decision is in the order of reciprocal of the square root of N (the number of data points).
>
> ***[ What are the advantages and disadvantages of the proposed methods and other existing MLC or XMLC methods?]***: As pointed out above, the proposed FrugalMCT is NOT a new MLC/XMLC method. Thus, comparison between it and other MLC/XMLC methods is not apples to apples. As the commercial ML APIs are fixed black-boxes, we could not run standard XMLC methods which typically involve changing the underlying model. Moreover, since there are multiple commercial ML APIs, the goal of FrugalMCT is to adaptively choose a different subset of APIs for different user queries, and aggregate their outcomes, instead of developing a new MLC.   FrugalMCT explicitly balances between accuracy and dollar cost of calling commercial APIs, which are underexplored in literature.
>
> ***[Can the proposed method converge?]***: Yes. As shown in Algorithm 1, the online API selector simply selects the API to maximize the difference between estimated accuracy and normalized cost (s^p(.) as defined in the 7th line of Section 3.3, page 5) at each iteration. This maximization only requires checking the objective for finite (K) choices and thus converges naturally.
>
> ***[Can you compare your method and other XMLC methods on XMLC datasets?]***: As we have clarified above, FrugalMCT is proposed to solve a new problem of adaptively selecting and combining commercial ML APIs, not solving XMLC problems. Moreover, standard XMLC approaches cannot be directly used to combine ML APIs, since we do not have whitebox access or the ability to retrain these APIs.  We have added a comparison to the previously published state-of-the-art FrugalML on the COCO dataset. Training FrugalML to predict 2^80 label combinations of COCO did not converge after 24 hours of training, while FrugalMCT’s entire training took less than one minute.

---

> ### Author Response · Authors · 2021-11-29
> **Dear Reviewer S7of: we'd love to hear if you have any further questions after our response**
>
> Dear Reviewer,
>
> Thank you very much for your time and feedback! We hope our response has answered your questions and you'd consider raising your score in light of it. In particular, several points you had asked about--runtime of the method and regret bound--were in the original paper and we highlighted the relevant figures and theorems in the response. Please let us know if you have any further questions and we are happy to follow up!
>
> Thank you again.

---

### Official Review · Reviewer_py8t · 2021-11-02

**Correctness:** 4
**Technical Novelty And Significance:** 3
**Empirical Novelty And Significance:** 4
**Recommendation:** 8
**Confidence:** 3

**Main Review:**

It is a practical problem to combine the existing ML APIs for better performance under a budget. This paper focus on three multi-label classification tasks, which is more challenging than single-label classification. The most related previous work FrugalML ignores correlation between different ML APIs, while FrugalMCT could select from the combination of them.

This paper provides sufficient formalism and proofs. The empirical results show that FrugalMCT is effective on three multi-label classification tasks, including computer vision and NLP tasks. And the GitHub model cost is also analyzed. The case study demonstrate that the proposed FrugalMCT improves the classification results compared to the single ML API.

The authors mention that the complexity of FrugalML is exponential to the number of labels. It could be interesting to compare the training and inference time of FrugalML and FrugalMCT with different ML APIs.

**Typo**

In the caption of Figure 1 *a cheap open source SSD model from GitHub **is is** high*.

**Summary Of The Paper:**

This paper addresses the practical task to use the combination of ML APIs for multi-label classification. Different from the related work FrugalML which ignores the correlation between ML APIs, the proposed FrugalMCT allows selecting and combining the different ML APIs based on a budget. Sufficient theoretical and empirical analyses are provided to demonstrate the effective of FrugalMCT.

**Summary Of The Review:**

This paper addresses a practical problem to combine the different ML APIs for multi-label classification and provides sufficient theoretical and empirical analyses. Recommended for acceptance.

---

> ### Author Response · Authors · 2021-11-18
> **Thank you for your helpful summary and support**
>
> Thank you for your helpful summary and support for the paper! We answer your questions as follows.
>
> ***[Compare the training and inference time of FrugalML and FrugalMCT]***: Let us take the performance on dataset COCO with budget 10 (same as the best commercial API) as an example. The training and inference time of FrugalMCT was 59.5s and 0.47s, respectively. Training FrugalML for all (2^80) possible label sets did not finish within 24 hours. Thus, we grouped all possible label sets into a few (80) single labels by K-means clustering, and found that FrugalML took 6627s for training and 0.43s for inference. In addition to fast training,  FrugalMCT also enjoys an accuracy (0.53) notably higher than FrugalML (0.51).
>
>
> ***[Typo]***: Thank you, we have fixed the typos in the uploaded revised paper..

---

### Official Review · Reviewer_R1P4 · 2021-11-02

**Correctness:** 4
**Technical Novelty And Significance:** 3
**Empirical Novelty And Significance:** 4
**Recommendation:** 8
**Confidence:** 4

**Main Review:**

# Soundness of the claims
The presented work is both theoretically grounded and empirically evaluated. In theory the authors' approach has convenient performance guarantees and can be implemented efficiently due to leveraging the specific structure of the problem. In the empirical evaluation the authors also prove the practical benefit of their approach demonstrating that their approach can achieve a higher predictive accuracy at a lower cost than the single best API, thereby outperforming the single best MLaaS platform in terms of both accuracy and cost which is not a really surprising result as long as the performances of the different APIs are complementary but still a strong result with practical benefits.

# Significance
The contribution is significant and I estimate that this work will have great impact on the respective community.

# Novelty
Although the basic idea of selecting APIs has already been done in previous work (FrugalML), except for the basic idea, the remaining paper is to the best of my knowledge novel.

# Relation with prior work
Related work is adequately and discussed in sufficient detail.

# Clarity of writing
The paper is very well structured and clearly written. As already said, I really enjoyed the reading.

# Relevance
The paper is definitely relevant to the machine learning community. The considered problem is relevant from a practical and very interesting from a methological perspective.

**Summary Of The Paper:**

In the paper "FrugalMCT: Efficient Online ML API Selection for Multi-Label Classification Tasks" the authors present an approach to learn how to select online APIs that machine learning as a service (MLaaS) for the problem of multi-label classification. As APIs differ in terms of accuracy and query cost, their approach achieves greater accuracy than a single API at a lower cost than the best single API.

**Summary Of The Review:**

Overall, I really enjoyed reading the paper and it was refreshing to see such a practical application and benefit for the end-user when using the approach in terms of both predictive accuracy and reduced costs. Therefore, I recommend to accept the paper.

---

> ### Author Response · Authors · 2021-11-18
> **Thank you for your helpful summary and support!**
>
> Thank you for your helpful summary and support for the paper!

---

### Official Review · Reviewer_rXPL · 2021-11-08

**Correctness:** 3
**Technical Novelty And Significance:** 3
**Empirical Novelty And Significance:** 3
**Recommendation:** 6
**Confidence:** 3

**Main Review:**

## Strengths:
- novel problem setup of MLaaS, which is very practical for industrial practionioners
- solving the API selection problem as relaxed integer programing seems techincal sounds
- most of the writing is clear and easy to follow

## Weakness:
- While the paper aims to solve for multi-label problem, most components in the proposed FrugalMCT seem not tailor for the mulit-label problem, except for label combination part mentioned in Sec 3.4.
- FrugalMCT needs call all candidate APIs do prediction on the training set so that it can collect supervision to learn the accuracy predictors. This step induces extra costs, but did not show up in the experiment results.

## Major Comments
- It seems to me that we can also apply FrugalMCT framework to an online production systems, where we have several ML predictors each with different inference latencies. We would like to select multiple ML predictors under the total latency constraint while striving for the best overal accuracy. It would be very interesting to see the impact of FrugalMCT in such cases.

## Minor Comments
- In the caption of Figure 1, "is is high" => "is high"


**Summary Of The Paper:**

Given several mutli-label machine learning APIs, this paper study how to select those APIs under a budget constrant while striving to improve the overall accuracy. The author first formulate the budget API select problem as an integer linear programming problem, then relax the integer contraint and solving the relax problem in dual. The advantage of such modeling is to have a fast decision function for online deployment of their API selection systems. The experiment results are promising and consists of several ablation studies.


**Summary Of The Review:**

This paper is novel and techinical sounds. The proposed framework is not specially optimized for multi-label problems, but it is still a good paper to see in ICLR. Thus, I am incline that is paper is marginally above the acceptance threshold.

---

> ### Author Response · Authors · 2021-11-18
> **Thank you for your helpful feedback and support**
>
> Thank you for your helpful feedback and support for the paper! We answer your questions below.
>
> ***[What is the cost of training a FrugalMCT strategy?]***: Both dollar cost and computation time  of training are usually much smaller than the cost of doing inference with ML APIs at scale afterward.This is because (i) training is a one-time cost and (ii) FrugalMCT requires a small number of label annotations (a few thousands, see Figure 5). Consider the image tagging task as an example: the dollar cost of calling all APIs is 0.0006+0.001+0.0015=0.0031 dollars per image. Labeling for (say) five thousands images takes 0.0031x5000=15.5 dollars. Training a FrugalMCT strategy on COCO takes 59.5s on a desktop. This is much cheaper than calling the selected APIs after at large scale (e.g., millions of images).  We have added more discussions of this to the Supplement of the revised paper.
>
> ***[Can FrugalMCT be applied for other resource-limited scenarios (e.g., latency-limited)?]***: Yes. The cost metric is general, and can be latency and energy assumption. We agree with you that it’d be very interesting to apply FrugalMCT to improve latency in future works.
>
> ***[typo]***: Thank you, we have fixed the typos in the uploaded revised paper.

---

### Author Response · Authors · 2021-11-18
**Thank you for the reviews**

We thank all the reviewers for their helpful feedback and support of the paper. We have uploaded a revised manuscript based on your suggestions.

---

### Decision · Program_Chairs · 2022-01-20

**Decision:**

Reject

**Comment:**

The authors study a practical problem of selecting/combining existing multi-label classification APIs under a budget constraint for a specific problem instance on hand. The task can be viewed as an (online) integer programming problem when given an accuracy estimator for the combination performance. The authors relax the integer constraints and propose a framework to solve the task in the dual form. They also run experiments to validate that the proposed framework is advantageous (cost or accuracy-wise) over the best single API.

Most of the reviewers are positive about the practical value and the potential impacts of the work in applications/products/services. There are several disputes between the authors and some reviewers that cannot be fully resolved during the rebuttal. In the end, no reviewers express willingness to strongly champion for the acceptance of the paper, making the paper a borderline case. The decision is based on a careful examination of the current manuscript and every side's opinions.

* Novelty: Some reviewers question about the novelty of the work. There are two aspects about novelty: one is on whether the problem itself is novel (are the authors trying to propose a new multi-label method?) In this aspect, the authors' response, which states that they are not aiming at proposing a new method, but at solving an automation task for MLaaS users, appears believable. The other aspect is whether the solution technique, namely the relaxed integer programming and other techniques, are sufficiently novel. Some reviewers find the novelty aspect satisfactory, while others believe that the proposed optimization technique have been widely used in machine learning community. The authors did not clarify the similarity/difference of the proposed technique to existing ones during the rebuttal. In this sense, the technical novelty is not well justified.

* Speed: Some reviewers are concerned about different aspects of the running time and other costs. The authors emphasized the rapid speed in inference phase, particularly in Figure 3. Less is discussed about the time needed for the training phase (although the authors claim to be much smaller than the inference time)---somehow even the most positive reviewers have some questions about this aspect. The authors could add more clarification about the different "time" costs to the discussion. One dispute between some reviewers and the authors is about the *complexity* analysis of time, which is indeed missing in the current manuscript and can be a nice-to-have for future todos.

* Theoretical Guarantee: One major dispute between some reviewers and the authors is on the theoretical guarantee provided. The reviewers suggest a regret-style bound, which compares the solution to the worst-case sequence; the authors provide an optimization-style bound, which compares the solution to the absolute optimal solution. Different bounds have their different roles for supporting the framework. Given that the authors have provided some reasonable bounds, the lack of regret bound is not taken against the authors.

* Specialty: One concern raised by some reviewers is that the technique does not seem particularly tailored for multi-label classification (except some minor parts). In this sense, it is nice to have for the authors to discuss more on the wider applicability of the technique, and/or include some more specialty of the multi-label classification problem into the technique design.

After taking all the factors above into account, and calibrating the received scores to the distribution across the papers, it seems that the paper could use some more revision before being mature enough as an impactful work.